# Clinico-Pathological Factors Determining Recurrence of Phyllodes Tumors of the Breast: The 25-Year Experience at a Tertiary Cancer Centre

**DOI:** 10.3390/jpm13050866

**Published:** 2023-05-21

**Authors:** Baijaeek Sain, Arnab Gupta, Aruni Ghose, Sudip Halder, Vishal Mukherjee, Samir Bhattacharya, Radha Raman Mondal, Aditya Narayan Sen, Bijan Saha, Shravasti Roy, Stergios Boussios

**Affiliations:** 1Department of Trauma & Orthopaedics, Imperial College London Healthcare NHS Trust, London W2 1NY, UK; 2Department of Surgical Oncology, Saroj Gupta Cancer Center and Research Institute, Kolkata 700063, India; 3Department of Medical Oncology, Mount Vernon Cancer Centre, East and North Hertfordshire NHS Trust, London SG1 4AB, UK; 4Department of Medical Oncology, Barts Cancer Centre, St. Bartholomew’s Hospital, Barts Health NHS Trust, London E1 1BB, UK; 5Department of Medical Oncology, Medway NHS Foundation Trust, Windmill Road, Gillingham ME7 5NY, UK; 6Department of Pathology, Saroj Gupta Cancer Centre and Research Institute, Kolkata 700001, India; 7Faculty of Life Sciences & Medicine, School of Cancer & Pharmaceutical Sciences, King’s College London, London SE1 9RT, UK; 8Kent Medway Medical School, University of Kent, Canterbury CT2 7LX, UK; 9AELIA Organization, 9th Km Thessaloniki—Thermi, 57001 Thessaloniki, Greece

**Keywords:** phyllodes, breast, recurrence, local, systemic, metastasis, surgery

## Abstract

Background: Phyllodes tumors (PTs) of the breast are rare fibroepithelial tumors that are generally more prone to recurrence. Aims and objectives: This study aimed to assess the clinicopathological features, diagnostic modalities, and therapeutic interventions, along with their respective outcomes, to identify the factors associated with a recurrence of PTs of the breast. Methodology: A retrospective cohort and observational study was conducted, which entailed analyzing the clinicopathological data of patients who were previously diagnosed or presented with PTs of the breast between 1996 and 2021. Data included the total number of patients diagnosed with PTs of the breast and their ages, tumor grade on initial biopsy, tumor location (left or right breast), tumor size, therapeutic interventions carried out (including surgery—either mastectomy or lumpectomy—and adjuvant radiotherapy), final tumor grade, recurrence status, type of recurrence, and time to recurrence. Results: We analyzed data on a total of 87 patients who were pathologically proven to have PTs, and 46 patients (52.87%) were found to have recurrences. All patients were female, with a mean age at diagnosis of 39 years (range 15–70). Patients aged <40 years had the highest incidence of recurrence, with a rate of 54.35% (n = 25/46), followed by patients aged >40 years, with a rate of recurrence of 45.65% (*n* = 21/46). A total of 55.4% of patients presented with primary PTs and 44.6% had recurrent PTs at presentation. The average time to local recurrence (LR) from the completion of treatment was 13.8 months, whereas for systemic recurrence (SR), it was 15.29 months. Surgery (mastectomy/lumpectomy) was the major determinant for local recurrence (*p* < 0.05). Conclusion: Patients who received adjuvant radiotherapy (RT) had a minimal recurrence of PTs. Patients who were found to have a malignant biopsy on initial diagnosis (triple assessment) had a higher incidence of PTs and were more prone to SR than LR. Surgery was a determining factor in the increased rate of LR, with lumpectomy associated with a higher incidence of LR than mastectomy.

## 1. Introduction

Phyllodes tumors (PTs) of the breast are uncommon fibroepithelial tumors that account for 0.3–0.5% and 2–3% of primary breast and fibroepithelial tumors, respectively. Although the prognosis of PTs is good, with a 10-year survival rate of 87%, they are generally more prone to recurrence [1]. The risk of recurrence depends on the tumor size and the surgical approach used. Most PTs are benign, with local recurrence (LR) occurring in a small proportion of cases. In certain cases, the tumor can metastasize, particularly in the context of malignant PTs, which carries with it a poor prognosis [2]. LR can occur in all PTs, with an overall rate of 21% and variations ranging from 10% to 17%, 14% to 25%, and 23% to 30% for benign, borderline, and malignant PTs, respectively. LR generally develops within 2 to 3 years. Distant metastases (DM) are almost exclusively a feature of malignant PTs. The lungs (66%), bones (28%), and brain (9%) are the most common sites of spread. In rare cases, metastases can affect the liver and heart [3]. We conducted this study to analyze the clinicopathological factors that determine the recurrence of breast PTs.

## 2. Aims and Objectives

This study aimed to assess the clinicopathological features, diagnostic modalities, and therapeutic interventions, along with their respective outcomes, to identify the factors associated with a recurrence of PTs of the breast.

## 3. Materials and Methods

### 3.1. Study Design and Participants

We carried out a retrospective cohort and observational study, which entailed analyzing the clinicopathological data of patients who were diagnosed or presented with PTs of the breast between 1996 and 2021 at Saroj Gupta Cancer Centre and Research Institute (SGCCRI), Kolkata. The center annually caters to approximately 50,000 cancer patients across the South-Asian subcontinent, primarily covering eastern India and the adjacent southern states, along with Bangladesh and a few other neighboring countries.

### 3.2. Data Collection and Analysis

After obtaining clearance from the Institutional Ethics Committee (IEC) of SGCCRI Kolkata, the following data were retrieved from medical records and analyzed—the total number of patients diagnosed with PT of the breast and their ages, tumor grade on initial presentation or biopsy, tumor location (left or right breast), tumor size, final tumor grade, therapeutic interventions carried out (surgery—either mastectomy or lumpectomy—and adjuvant radiotherapy), recurrence status, type of recurrence, and time to recurrence.

### 3.3. Statistical Methods

Data management and statistical analysis were performed using Python 3.0. Numerical (Python Software Foundation, Amsterdam, The Netherlands) data were summarized using means and standard deviations or median ranges. Categorical data were summarized as percentages. The associations among the categorical data were examined using the Chi-square test. All tests were conducted at a two-sided significance level of 0.05, with no adjustments made for multiple comparisons. We assessed all study variables to analyze the correlations with recurrence using univariate logistic regression, initially considering all clinicopathological factors, followed by multiple logistic regression analysis. The correlations of local and systemic recurrences with all clinicopathological variables were further analyzed using multinominal logistic regression.

## 4. Results

We analyzed data on 87 patients who were pathologically proven to have PTs of the breast and were treated between 1996 and 2021. Forty-six out of the 87 patients were found to have recurrences.

### 4.1. Demographic and Clinical Characteristics

All patients were female, with a mean age at diagnosis of 39 years (range 15–70). Patients aged <40 years had the highest incidence of recurrence, with a rate of 54.35% (*n =* 25/46), followed by patients aged >40 years, with a rate of recurrence of 45.65% (*n =* 21/46) (Figure 1). All patients presented with a breast mass. In 46 cases, the tumor was huge, encompassing almost the entire breast. A total of 34.78% (*n =* 16/46) of patients with a tumor size of > 10cm and 65.22% (*n =* 30/46) of patients with a tumor size of <10 cm showed recurrences. The average diameter of tumors at the time of presentation was found to be 8.65 cm. The largest diameter of a tumor at presentation was 27 cm, whereas the smallest was 3 cm. A total of 69.57% (*n =* 32/55) of right-sided tumors and 30.43% (*n =* 14/32) of left-sided tumors recurred. None of the patients had a family history of breast cancer or PT.

### 4.2. Local and Systemic Recurrence

Forty-eight out of 87 patients (55.17%) presented with primary PTs, and 39 patients (44.83%) had recurrent PTs at presentation. Among the patients who presented with recurrent PTs at presentation, 18 were diagnosed at the local stage and 21 had systemic metastasis. The most common type of recurrent PT observed in the final biopsy was malignant, accounting for 82.61% (*n =* 38/46) of cases, followed by borderline PTs, with benign PTs being the least frequently observed. A central pathological review was conducted for the final tumor grade. Out of the total cohort of 87 patients (52.87%) in this study, 46 had recurrences. The distributions of LR and SR among patients who underwent surgery and received adjuvant radiotherapy (RT) are shown in Figure 2. The longest duration to present as LR from the completion of treatment was found to be 84 months, whereas the shortest was 2 months. The longest and shortest durations to present as SR from the completion of treatment were 96 months and 1 month, respectively. The average duration to present as LR from the completion of treatment was 13.8 months, whereas for SR, it was 15.29 months (Figure 3).

Several factors were studied for their correlation with LR and SR, as shown in Table 1 and Table 2. The most important factor was found to be surgery (mastectomy/lumpectomy), with *p* = 0.0067 in the univariate analysis, *p* = 0.008 in the logistic regression, and *p* = 0.007 in the multinomial logistic regression when associated with LR. Adjuvant RT was found to be a significant factor for SR, along with the initial histological subtype of the tumor, as benign cases showed lower recurrence rates (15.22%) compared with malignant PT cases (84.78%). DM occurred in *n =* 21 patients (24.13%, *n =* 21/87), 8 with borderline PT and 13 with malignant PT. The most common site for metastasis was the lungs. Other areas for metastasis included the bones, iliac nodes, chest wall, and mediastinal nodes.

## 5. Discussion

PTs were first described by Müller in 1838 as Cystosarcoma phyllodes. The word phyllodes originates from the Latin word ‘phyllodium’, meaning leaf-like, and it is based on a gross pathological description of a laminar, voluminous, cystic, and fleshy tumor of the breast [1]. The World Health Organization (WHO) has classified PTs histologically as benign, borderline, and malignant. PTs can be found in all age groups. However, the average age at presentation is 45 years [2]. A triple assessment, which includes clinical, radiological, and histological examinations, is used to diagnose PTs. PTs can mimic a fibroadenoma in terms of clinical appearance. Breast imaging of PTs can also resemble fibroadenomas (FAs). The cytologic diagnosis of PTs through biopsy is usually considered unreliable. However, a punch needle biopsy is superior to a fine needle puncture [3]. In our study, mammography was primarily used as the imaging modality, as CT or MRI were less accessible due to the lower socioeconomic profile of our cohort. Regarding histology, fine needle aspiration cytology (FNAC) was performed first, followed by an excisional biopsy of the surgical specimens.

PTs primarily occur in women, although there have been reports of a few cases in men, all of which have been associated with gynecomastia. Latino women and East Asians born in Central or South America who live in the United States are at higher risk. The median age at presentation of PT is 45 years, with an age range of 9 to 93 years. Genetic mutations in chromosomal regions +1q, +5p, +7, +8, 9p, 10p, 6, and 13 are associated with borderline and malignant PTs of the breast. Women with Li–Fraumeni syndrome have an increased risk of PT [4].

Unlike breast cancer, PTs begin outside the lobules and ducts in the breast’s connective tissue called the stroma, including the ligaments and fatty tissue that surround the lobules, ducts, lymph, and blood vessels. In addition to epithelial cells from the ducts and lobules, PTs can also contain stromal cells. They most likely develop de novo, although there have been reports of progression from FA to PT. PTs can occur due to growth factors produced by the breast epithelium. Trauma, pregnancy, increased estrogen activity, and lactation are sometimes considered factors that can stimulate tumor growth. Endothelin-1, a stimulator of breast fibroblast growth, could be a contributing factor [5]. PTs usually manifest as unilateral, firm, enlarging, painless breast masses that stretch the overlying skin and are associated with marked distension of the superficial veins in the upper outer quadrant of the breast. They are rarely bilateral, accounting for about 1.8% of cases. In some patients, the lesion may appear after many years of rapid growth. In rare cases, PTs can exhibit blue discoloration, dilated skin veins, skin ulcers, nipple retraction, and palpable axillary lymph nodes [6].

A PT is a rare breast lesion that usually occurs in women aged 35–50 years. However, malignant PTs may appear later [7]. This correlates with our cohort, where the median age at diagnosis was 39 years compared to 46 years for malignant PTs. Although these tumors have an average size of 5 cm, lesions of up to 40 cm have been reported [7]. The relationship between tumor size and malignancy is controversial; however, rapid growth has been observed in malignant tumors [8]. In our study, the mean pathologic tumor size was 8.65 cm, which is consistent with the existing literature, and the largest PT was 27 cm in diameter, which was diagnosed as a malignant PT. Another unclear relationship exists between tumor size and recurrence rates. Kim et al. postulated that smaller tumors (size ≤ 4 cm) may receive less aggressive surgical treatment, which could potentially lead to higher recurrence rates [9].

In our cohort, 34.7% of patients with a tumor size > 10 cm and 65.2% with a tumor size < 10 cm had a recurrence. In addition, the smallest tumor was found to be 3 cm in diameter. A tumor size of 10 cm was considered to be the median value within the interquartile range. Compared to the other studies cited, our cohort had relatively larger tumor sizes [7,8,9]. This may be because the majority of our population came from a lower socioeconomic background, and between 1996 and 2021, there was a significant lack of awareness and knowledge about breast self-examination (SBE) and the early detection of breast tumors. Therefore, these patients often presented with larger tumor sizes by the time they became aware of their condition. These results are discussed in the following sections.

The prevalence of PTs in the right and left breasts is comparatively similar. In addition, multifocality and bilaterality are rare in PTs [10]. In our cohort, the right breast was more affected than the left breast (55 and 32 cases, respectively).

FNAC, core needle biopsy (CNB), and incision and excision biopsies can be used in the preoperative histopathologic diagnosis of PTs. Distinguishing between benign PTs and FA and malignant PTs from metaplastic spindle cell carcinoma and primary breast sarcoma is one of the greatest histopathological dilemmas [11]. CNB is considered more reliable than FNAC in making a correct diagnosis because it can provide specific histopathological findings. However, a sensitivity of about 65% has been reported for the definitive diagnosis of PTs [12]. Eleven patients were initially misdiagnosed as benign by FNAC and later diagnosed as malignant by the final biopsy, whereas only one patient was correctly diagnosed as malignant by FNAC. Seventy-five patients were correctly diagnosed as malignant by CNB. In this study, the histological diagnosis was correctly confirmed by CNB in most cases, with a diagnostic accuracy of approximately 85.76% for PTs. This is consistent with other studies that found CNB to be a valuable tool in the differential diagnosis of PTs due to its high specificity and sensitivity [13].

The association of recurrences consistently demonstrated similar results, where the final biopsies of malignant and borderline tumors by CNB revealed recurrences of 52.78% and 72.72%, respectively. In our cohort, CNB played an important role in the preoperative histopathologic diagnosis of PTs, whereas FNAC was used in 16 patients, with only 1 case suspected of having a PT. Excisional biopsy was preferred in cases where FA or another benign lesion was strongly suggested in the preoperative clinical and radiographic evaluations.

Widespread local excision (WLE), which involves removing the tumor while maintaining a minimum 1 cm free microscopic margin, is the main method suggested for PTs; however, patients with large malignant tumors or those with a high tumor-to-breast tissue ratio may require a mastectomy [14,15]. However, in a systematic review of 12 studies involving >1700 patients by Shaban et al., there was no difference in recurrence rates between a 10 mm margin (7.9%) and a 1 mm margin (5.7%) (*p* = 0.124). An increase in recurrence rates (12.9%) (*p* = 0.006) [16] was found when the edge of the focus was affected (presence of tumor cells). In our study, WLE was the most commonly performed primary surgery (44.1%). In patients with recurrences after WLE, surgical management included mastectomy (61.4%). Axillary dissection is not recommended as part of routine surgical management because PTs usually spread through a hematogenous route and nodal involvement is extremely rare. However, axillary dissection may be considered in patients with malignant PTs and axillary metastases [17]. In the 16 cases of benign PTs treated using WLE, 3 recurrences occurred, 2 of which progressed, 1 to a borderline and the other to a malignant PT. Of the four benign cases treated with mastectomies, only one recurred. Another study by the same researchers included a series of 33 cases. They found no association between the width of the surgical margin and disease recurrence [18,19].

A retrospective review of 44 Asian cases found no cases of LR in benign tumors treated with simple excision (enucleation) regardless of marginal status after a mean follow-up of 47.6 months. Therefore, a benign PT diagnosed after a representative excisional specimen collection can be treated conservatively, even if positive margins are observed [20]. Across our cohort, four patients initially diagnosed with benign PTs eventually developed local recurrence (LR) after WLE. However, the initial diagnosis was based on FNAC. The final biopsy after resection revealed that two of these patients were borderline and two were malignant. LR occurred in both borderline and malignant tumors due to inconclusive findings in the FNAC and final biopsy, demonstrating the malignant/borderline nature of the tumors.

Conversely, malignant PTs are associated with a 29.6% recurrence rate, with metastases and death observed in 22% of affected patients, underscoring the need to identify this subset of aggressive PTs for complete surgical eradication [21]. In our study, those treated with a mastectomy had better outcomes and fewer recurrences for malignant PTs. The number was even lower for those who received adjuvant radiation therapy. These results are similar to those reported in the literature [22]. DM typically occurs in 10% of cases and most commonly affects the lungs and bones [23]. In our cohort, DM occurred in 21 patients (24.13%), 8 with borderline PTs and 13 with malignant PTs. The most common site of metastasis was the lungs. Other areas of metastasis included the bones, pelvic nodes, chest wall, and mediastinal nodes.

In our study, most patients with malignant PTs were offered adjuvant RT. The 33 patients who received adjuvant radiotherapy after mastectomy showed no recurrence (Figure 2). From this, we can conclude that adjuvant RT significantly reduced recurrence rates regardless of the final biopsy type. Against this background, future studies can examine the treatment options for the three subgroups, namely benign, borderline, and malignant PT. There is no global consensus on the role of adjuvant RT and chemotherapy in the management of PTs [24]. However, the application of RT to the chest after surgery in borderline and malignant PTs has been shown to reduce the risk of LR [25]. In our study, the mean RT dose was 50 Gy/25 fractions, followed by tumor bed enhancement in cases of WLE.

Adjuvant RT should be considered on an individual basis in patients with borderline and malignant PTs. However, clear guidelines are still lacking, necessitating further studies [11]. In a study by Ibreaheem et al., adjuvant RT was administered to nine patients, two with borderline PTs and seven with malignant PTs. However, after radiation therapy, five of these patients relapsed, all of whom had malignant PTs [26]. Kim et al. used a large population database known as SEER (Surveillance, Epidemiology, and End Results Program) to investigate the impact of the use of adjuvant RT in malignant PTs on cancer-specific survival (CSS). Of 1974 patients, 825 had a mastectomy and the remaining 1149 had a lumpectomy. Subsequently, 130/825 mastectomy patients and 122/1149 lumpectomy patients received adjuvant RT. When taking into account adverse risk factors, such as large tumor size and high grade, the use of RT did not provide a significant benefit in terms of CSS compared to non-RT modalities [27].

The retrospective nature of our study is a limitation. Due to the age of the medical records, we could not calculate survival rates. Therefore, it was difficult to predict the disease-free survival (DFS) rate in patients with recurrent tumors after treatment. In addition, the use of FNAC in diagnosing PTs remains controversial as it relies on taking an adequate and representative sample. Sampling issues are also caused by the heterogeneity of PTs.

In our entire cohort, local recurrence was found in a total of four patients initially diagnosed with benign phyllodes tumors after extensive local excision/lumpectomy. However, the initial diagnosis was based on FNAC. The final biopsy after resection revealed that of these four patients, two were borderline and two were malignant. Based on the inconclusive findings of the FNAC and final biopsy, which demonstrated the malignant/borderline nature of the tumors, we confirmed that LR occurred in both borderline and malignant tumors. We also concluded that patients who received adjuvant radiation had a significantly lower rate of recurrence, regardless of their final biopsy type. However, in this cohort, most patients with malignancies were offered adjuvant RT. In order to reach definitive conclusions, future controlled trial studies focusing on the three subgroups are necessary, where we can determine and demonstrate the results individually for benign, borderline, and malignant lesions.

The tumor sizes in our patient cohort were typically large, with the smallest being 3 cm in diameter. Based on the sample size and biostatistical analysis performed in this cohort, a tumor size of 10 cm was considered the mean value within the interquartile range observed. Different types of phyllodes tumors may exhibit different tumor sizes. However, we concluded that the main reason the patients in our cohort had larger tumors was that the study cohort consisted of many patients who presented to our hospital between 1996 and 2021 who were only from the US Indian subcontinent. In this regard, breast self-examination (BSE) techniques and women’s health awareness for the early detection of cancer were completely unknown until the early 2020s. The literacy levels of the patients varied, as people of all backgrounds visit our hospital, regardless of their financial background. Therefore, we hypothesized that many of them from lower socioeconomic backgrounds probably had very limited access to and knowledge about breast cancer screening. Even in today’s clinical practice within the Indian Diaspora, rare and common breast lesions are undetectable and unknown to many, and by the time women become aware of them, they are often in advanced stages, leading to a poor prognosis and limited survival rates. The related treatment results were examined based on the individual characteristics of the patients and their respective treatments following the assessment of the tumor after the preliminary examination.

The genetic risk factors for phyllodes tumors are largely unknown; however, phyllodes tumors in Li–Fraumeni syndrome patients and a mother–daughter pair have been described in the literature. Rare cases of phyllodes tumors in men are often associated with gynecomastia, suggesting a potential role of hormonal imbalance in their development. Researchers have postulated that stromal induction of phyllodes tumors may be due to growth factors produced by the mammary epithelium. Trauma, pregnancy, increased estrogen activity, and lactation are sometimes considered factors that stimulate tumor growth. The nature of these factors is not fully understood, but endothelin-1, a stimulator of breast fibroblast growth, may be a contributing factor.

Abnormal L1 retrotransposition events in tumors with high rates of retrotransposition can result in significant reorganization of the cancer genome. L1-mediated deletions can promote the loss of megabase-sized regions of a chromosome, which may involve centromeres and telomeres. Most of these genomic changes would be detrimental to a cancer clone. However, L1-mediated deletions can promote cancer-initiating rearrangements that involve a loss of tumor suppressor genes and/or enhancement of oncogenes. Through this mechanism, cancer clones acquire new mutations that help them survive. The *BRCA1* and *BRCA2* genes encode proteins required for homologous recombination (HR) to repair broken DNA. If mutations inactivate either the *BRCA1* or *BRCA2* gene, the broken DNA can become pathogenic. Pieces of DNA are lost or reattached in the wrong places on the original or other chromosomes [28]. In BRCA1 or BRCA2 mutants, these errors result in chromosomal rearrangements and shifts that are characteristic of hereditary breast cancer [29]. Chromosomal rearrangements can be critical events leading to hereditary breast cancer, but our knowledge of the causes of these events is limited. Immunodeficiencies in BRCA1 and BRCA2 mutants may allow for the reactivation of latent EBV infections or new herpesvirus infections. DNA breaks caused by exogenous nucleases of human herpesvirus 4 (EBV) can then become pathogenic.

The link between breaks in breast cancer chromosomes and viruses may be actionable. Based on the available evidence, germline genetic testing should be offered to all women diagnosed with breast and epithelial ovarian cancers [30]. The analysis should be able to detect deleterious variants in all HR genes, not just the *BRCA1* and *BRCA2* genes. Moreover, tumor testing, at least for the *BRCA1* and *BRCA2* genes, is recommended for all women who test negative for germline pathogenic variants. Poly (ADP-ribose) polymerase (PARP) inhibitors have garnered much attention and exemplify a paradigm of “bench-to-bedside” medicine. HR deficiency remains a strong predictor of the clinical utility of these agents. The current state of HR deficiency testing can identify patients with breast, epithelial ovarian, and prostate cancers who are most likely to benefit from PARP inhibitors. Precise biomarkers for predicting negative reactions are crucial. A better understanding of *BRCA* genes and their role in both cancer development and outcomes offers great potential to prevent many cases through improved access to genetic screening and to revolutionize long-term treatment [31].

## 6. Conclusions

Surgery and adjuvant RT contributed significantly to the recurrence of PTs, where patients who underwent only lumpectomies had a higher recurrence than patients who underwent mastectomies. Patients who received adjuvant RT had a minimal recurrence of PTs. Patients who were found to have a malignant biopsy on initial diagnosis (triple assessment) had a higher incidence and were more prone to SR than LR. Surgery was also a determining factor in the increased rate of LR (lumpectomy > mastectomy).

## Figures and Tables

**Figure 1 jpm-13-00866-f001:**
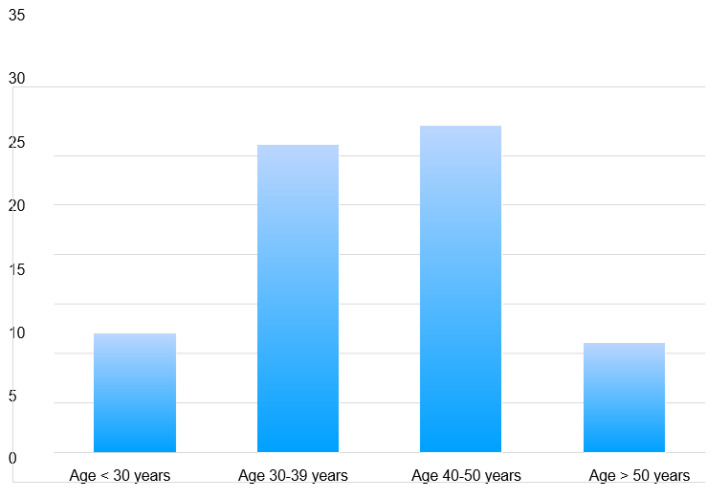
Incidence of phyllodes tumors of the breast by age and decade in the study cohort.

**Figure 2 jpm-13-00866-f002:**
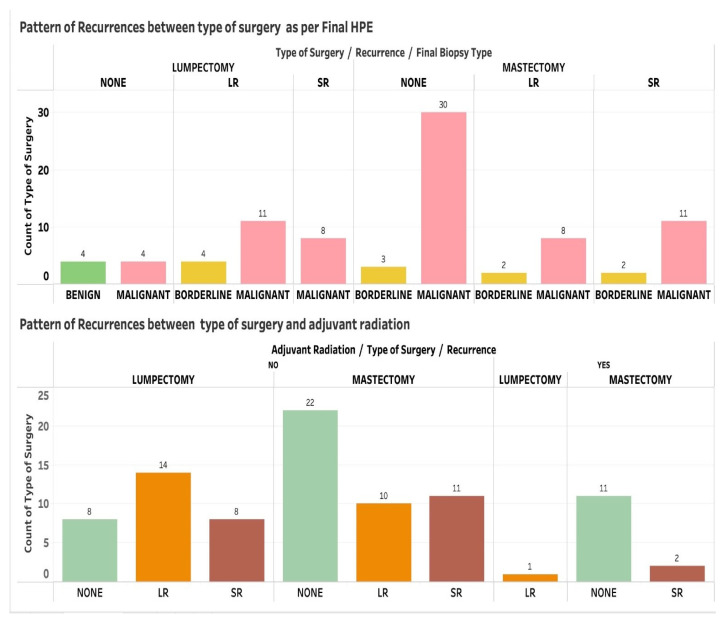
The distributions of local and systemic recurrence among patients who underwent surgery and received adjuvant radiotherapy. LR = local recurrence; SR= systemic recurrence.

**Figure 3 jpm-13-00866-f003:**
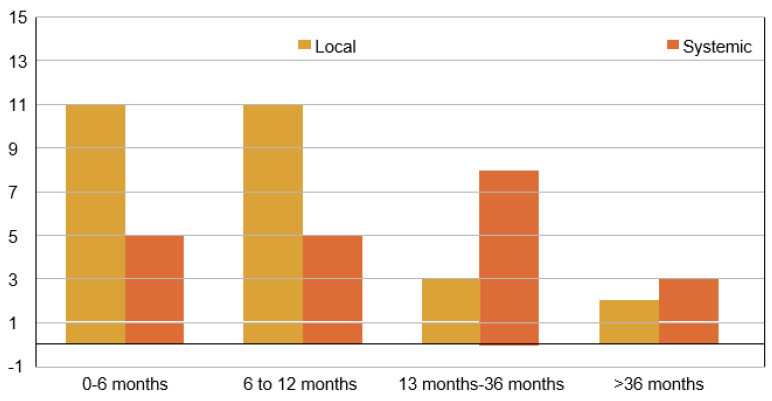
Time to recurrence following completion of treatment.

**Table 1 jpm-13-00866-t001:** Relationship between different clinicopathological factors and recurrent PTs of the breast.

VARIABLES		Recurrence YES		Recurrence NO		*p* Value (Univariate Analysis)	*p* Value (Multivariate Logistic Regression Analysis)
		N	%	N	%		
Biopsy	BENIGN	07	15.22	09	21.95	0.956	0.160
	MALIGNANT	39	84.78	32	78.05		
	YES	21	45.65	22	53.66		
Age > 40 years	NO	25	54.35	19	46.34	0.967	0.398
	BENIGN	0	0.00	4	9.76		
	BODERLINE	08	17.39	03	7.32		
Final Tumor grade	MALIGNANT	38	82.61	34	82.93	0.398	0.093
	LEFT	14	30.43	18	43.90		
Side	RIGHT	32	69.57	23	56.10	0.792	0.444
	LUMPECTOMY	23	50.00	08	19.51		
Surgery	MASTECTOMY	23	50.00	33	80.49	0.0067	0.008
	YES	03	6.52	11	26.83		
Adjuvant Radiotherapy	NO	43	93.48	30	73.17	0.157	0.041
	YES	16	34.78	15	36.59		
Size > 10cm	NO	30	65.22	26	63.41	0.999	0.766

**Table 2 jpm-13-00866-t002:** Relationship between different clinicopathological factors and local vs. systemic recurrence using multinomial logistic regression analysis.

VARIABLES		Recurrence YES		Recurrence NO		*p* Value (Local Recurrence)	*p* Value (Systemic Recurrence)
		N	%	N	%		
	BENIGN	07	15.22	09	21.95		
Biopsy	MALIGNANT	39	84.78	32	78.05	0.808	0.049
	YES	21	45.65	22	53.66		
Age > 40 years	NO	25	54.35	19	46.34	0.294	0.206
	BENIGN	0	0.00	4	9.76		
	BODERLINE	08	17.39	03	7.32		
Final Tumor grade	MALIGNANT	38	82.61	34	82.93	0.188	0.478
	LEFT	14	30.43	18	43.90		
Side	RIGHT	32	69.57	23	56.10	0.278	0.951
	LUMPECTOMY	23	50.00	08	19.51		
Surgery	MASTECTOMY	23	50.00	33	80.49	0.007	0.083
	YES	03	6.52	11	26.83		
Adjuvant Radiotherapy	NO	43	93.48	30	73.17	0.097	0.126
	YES	16	34.78	15	36.59		
Size > 10 cm	NO	30	65.22	26	63.41	0.917	0.425

## Data Availability

The datasets generated and/or analyzed during the current study are not publicly available due to ethical considerations but are available from the corresponding author on reasonable request.

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
