# Peer review of "Clinico-Pathological Factors Determining Recurrence of Phyllodes Tumors of the Breast: The 25-Year Experience at a Tertiary Cancer Centre"

_jpm, 2023, doi:10.3390/jpm13050866_

Round 1

Reviewer 1 Report

Authors should report the quality of surgery and describe more the diagnostic procedure (type of imaging, biopsy etc...)

Authors report a small retrospective cohorte of phylode tumor.

The reccurence rate is higher than previously report, probably because of the size of PT at diagnostic in this serie. Moreover, diagnostic procedure (imaging, MRI biopsy...), and surgery margin are not reported.

Very large serie of PT have been previously describe, I think the relevance for the medical communoty of this work is very low because results cannot be transpose in all country.

Author Response

Dear Editor and Reviewers,

I am pleased to resubmit for publication the revised version of the manuscript entitled “Clinico-pathological factors determining recurrence in Phyllodes tumour of the breast: The 25-year experience at a tertiary cancer centre”.

Thankfully the reviewers provided us with a great deal of guidance, regarding how to better position the article. We are hopeful you agree that this revision will update our comprehensive review. All the comments have been addressed, as shown in the revised version of the manuscript, along with this point-by-point response to the reviewers' comments.

All corresponding are blue changes in the manuscript.

REVIEWER 1

"Authors should report the quality of surgery and describe more the diagnostic procedure (type of imaging, biopsy etc...)

Authors report a small retrospective cohorte of phylode tumor.

The reccurence rate is higher than previously report, probably because of the size of PT at diagnostic in this serie. Moreover, diagnostic procedure (imaging, MRI biopsy...), and surgery margin are not reported.

Very large series of PT have been previously describe, I think the relevance for the medical communoty of this work is very low because results cannot be transpose in all country."

Response:

Quality of surgery: Hopefully, we have adequately covered that in the discussion section.

Type of diagnostic procedure: “In our study, the principal imaging modality used was mammography as CT or MRI was less accessible owing to the lower socioeconomic profile of our cohort. In terms of histology, fine needle aspiration cytology (FNAC) was initially performed followed by excision biopsy of the surgical specimens.” (First paragraph in the discussion section).

Thank you for your opinion. From our point of view, even though the study is retrospective and very large series of PT have been previously published, real world data in rare clinical entities is worthy to be reported in the literature.

Reviewer 2 Report

Thank you for your work on phylloides tumours.  However, could you please check your spelling and see that it is consistent throughout the manuscript.  Occasionally, the authors use 'phylloides' and at other times, 'phyllodes'.  Please be consistent as to whether you choose to use the British or American spelling.  

There are also typesetting issues where the space between two words are missing.  Please check.

Kindly clarify the association between recurrences and type of surgery.  Were there recurrences among women who were diagnosed with benign phylloides tumours and underwent wide excision?  Or did they only occur among those with borderline and malignant tumours?  Were these recurrences significantly reduced with radiotherapy?  If so, would there be differential treatment recommendations for benign, borderline and malignant lesions?  

The tumour size in your cohort of patients tend to be large, with the smallest being reported to be 3 cm.  There are other series, which were cited by the authors, where patients presented with smaller tumours.  Please comment and offer reasons why your cohort of patients presented with larger tumours and what impact this may have on treatment outcomes.

Please clarify this apparently contradictory statement: 'In our study, WLE was the most commonly performed initial surgery (44.1%), while mastectomy was mostly done in recurrent cases (61.4%).

Author Response

Dear Editor and Reviewers,

I am pleased to resubmit for publication the revised version of the manuscript entitled “Clinico-pathological factors determining recurrence in Phyllodes tumour of the breast: The 25-year experience at a tertiary cancer centre”.

Thankfully the reviewers provided us with a great deal of guidance, regarding how to better position the article. We are hopeful you agree that this revision will update our comprehensive review. All the comments have been addressed, as shown in the revised version of the manuscript, along with this point-by-point response to the reviewers' comments.

All corresponding are blue changes in the manuscript.

REVIEWER 2

"Thank you for your work on phylloides tumours. However, could you please check your spelling and see that it is consistent throughout the manuscript. Occasionally, the authors use 'phylloides' and at other times, 'phyllodes'. Please be consistent as to whether you choose to use the British or American spelling.

Response: These have been amended and changes highlighted in blue.

There are also typesetting issues where the space between two words are missing. Please check.

Response: These have been amended.

Kindly clarify the association between recurrences and type of surgery. Were there recurrences among women who were diagnosed with benign phylloides tumours and underwent wide excision? Or did they only occur among those with borderline and malignant tumours? Were these recurrences significantly reduced with radiotherapy? If so, would there be differential treatment recommendations for benign, borderline and malignant lesions?

Response: Figure 2 describes in detail regarding recurrence and types of surgery. Both local and systemic recurrence have been segregated in relation to surgery and adjuvant radiotherapy. We have modified the figure description to – “The distribution of local and systemic recurrence with patients having surgery and receiving adjuvant radiotherapy. (LR= local recurrence and SR= Systemic recurrence).”

Page 14 = “In our entire cohort, 4 patients who were initially diagnosed with benign PT were eventually found to have local recurrence (LR) following WLE. However, the initial diagnosis was based on FNAC. The final biopsy post resection revealed that out of 4, 2 were borderline and 2 were malignant. Hence, courtesy inconclusive FNAC and final biopsy proven malignant/borderline nature, LR occurred among borderline and malignant tumours.” (Eighth paragraph in the discussion section).

Page 14 = “In our study, adjuvant RT was offered to majority of malignant PT. 33 patients who received adjuvant radiotherapy following mastectomy showed no recurrence [Figure 2]. Hence, we can conclude that adjuvant RT significantly reduced recurrence rates irrespective of final biopsy type. Considering this, further trials with three subgroups, i.e., benign, borderline, and malignant PT, can be investigated for treatment recommendations.”(Ninth paragraph in the discussion section).

The tumour size in your cohort of patients tend to be large, with the smallest being reported to be 3 cm. There are other series, which were cited by the authors, where patients presented with smaller tumours. Please comment and offer reasons why your cohort of patients presented with larger tumours and what impact this may have on treatment outcomes.

Response:

Page 12 = “A tumour size of 10cm was considered the median figure within the interquartile range. Compared to the other studies cited, our cohort has a relatively larger tumour size [7-9]. This is because majority of our population originated from a lower socio-economic profile and presented during 1996 to 2021 i.e., circumstances implicating a significant lack of awareness or knowledge about self-breast examination (SBE) and early detection of breast tumours. Henceforth, by the time they would develop awareness, it would already have been a late tumour size presentation. Outcomes of such a presentation have been discussed in the following sections.”(Fourth paragraph in the discussion section).

Please clarify this apparently contradictory statement: 'In our study, WLE was the most commonly performed initial surgery (44.1%), while mastectomy was mostly done in recurrent cases (61.4%)."

Response: These have been modified to – “In our study, WLE was the most performed initial surgery (44.1%). In patients with recurrence following WLE, the surgical management done for them involved mastectomy (61.4%).” (Seventh paragraph in the discussion section).

Reviewer 3 Report

-Check english and typo errors.

-page 12: "Wide local excision (WLE), with removal of tumor keeping at

least 1 cm clear microscopic margins is the mainstay for PT while

mastectomy may be needed in patients with large malignant tumors

or those with high tumor-breast tissue ratio [14-15]." Personally I don't completely agree. 1cm free margin is not the mainstay any more. Please read this reference and include it in the manuscript: Shaaban M, Barthelmes L. Benign phyllodes tumours of the breast: (Over) treatment of margins - A literature review. Europ J clin Oncol, 2017 Jul;43(7):1186-1190.

page 13: "Adjuvant radiotherapy should be considered in patients with borderline and malignant PT". Int. guideline do not supprt use of RT for borderline tumors. Please read this reference and include it in the manuscript: Kim YJ , Kim K. Radiation therapy for malignant phyllodes tumor of the breast: An analysis of SEER data. Breast 2017 Apr (32):26-32.

Author Response

Dear Editor and Reviewers,

I am pleased to resubmit for publication the revised version of the manuscript entitled “Clinico-pathological factors determining recurrence in Phyllodes tumour of the breast: The 25-year experience at a tertiary cancer centre”.

Thankfully the reviewers provided us with a great deal of guidance, regarding how to better position the article. We are hopeful you agree that this revision will update our comprehensive review. All the comments have been addressed, as shown in the revised version of the manuscript, along with this point-by-point response to the reviewers' comments.

All corresponding are blue changes in the manuscript.

REVIEWER 3

"-Check english and typo errors.

-page 12: "Wide local excision (WLE), with removal of tumor keeping at

least 1 cm clear microscopic margins is the mainstay for PT while

mastectomy may be needed in patients with large malignant tumors

or those with high tumor-breast tissue ratio [14-15]." Personally I don't completely agree. 1cm free margin is not the mainstay any more. Please read this reference and include it in the manuscript: Shaaban M, Barthelmes L. Benign phyllodes tumours of the breast: (Over) treatment of margins - A literature review. Europ J clin Oncol, 2017 Jul;43(7):1186-1190.

Response:

However, Shaban et al’s systematic review of 12 studies with >1700 patients showed no difference in recurrence rates between a 10mm margin (7.9%) vs 1mm margin (5.7%) (p=0.124). Recurrence rates increases were noted in the event of focal margin involvement (presence of tumour cells) (12.9%) (p=0.006) [16].” (Seventh paragraph in the discussion section).

16. Shaaban, M., Barthelmes, L. Benign phyllodes tumours of the breast: (Over) treatment of margins - A literature review. Eur J Surg Oncol. 2017, 43(7), 1186-1190.

page 13: "Adjuvant radiotherapy should be considered in patients with borderline and malignant PT". Int. guideline do not support use of RT for borderline tumours. Please read this reference and include it in the manuscript: Kim YJ , Kim K. Radiation therapy for malignant phyllodes tumor of the breast: An analysis of SEER data. Breast 2017 Apr (32):26-32."

Response:

Kim et al used the large SEER (Surveillance, Epidemiology and End Results Program) population database to study the impact of adjuvant RT in malignant PT regarding cancer specific survival (CSS). Out of 1974 patients, 825 had mastectomy and remaining 1149 underwent lumpectomy. Following this, 130/825 mastectomy and 122/1149 lumpectomy patients had adjuvant RT considering adverse risk factors including large tumour size and high grade. Regardless of type of surgery, RT added nil benefit to CSS when compared to non-RT modalities [27].” (Prefinal paragraph in the discussion section).

27. Kim, Y.J.; Kim, K. Radiation therapy for malignant phyllodes tumor of the breast: An analysis of SEER data. Breast. 2017, 32, 26-32.

Round 2

Reviewer 1 Report

Ok

Reviewer 3 Report

now suitable for publication